# Effects of Genetic Mutation Sites in ADR Genes on Modern Chickens Produced and Domesticated by Artificial Selection

**DOI:** 10.3390/biology12020169

**Published:** 2023-01-20

**Authors:** Tomoyoshi Komiyama

**Affiliations:** Department of Clinical Pharmacology, Tokai University School of Medicine, Isehara 259-1193, Kanagawa, Japan; komiyama@tokai-u.jp; Tel.: +81-463-93-1121

**Keywords:** adrenergic receptor, domesticated chicken, artificial selection, cockfighting, neurotransmitter, nucleotide differentiation

## Abstract

**Simple Summary:**

Nine types of complete adrenergic receptor (ADR) gene sequences were analyzed, wherein twenty-four specific mutation sites were found as a result of the artificial selection of Shaver Brown and Shamo chickens. From the analysis, Shamo- and Shaver Brown-specific mutations, and those common to both breeds, could be separated into three groups. The results confirmed that eight mutation sites may be affected by artificial selection. Furthermore, the evolutionary analysis revealed that the identified mutations were not ancestral. These results confirmed that the eight mutations at these sites were artificially selected by domestication and breed specificity. *N_ST_* population analysis confirmed that there is a difference in the degree of genetic differentiation between the two populations. In particular, the *N_ST_* rate of *ADRA1D* (0.064) was most affected by artificial selection. This suggests that these mutations may exert different effects on vasoconstriction, smooth muscles, and the action of the digestive system relevant to the breed’s specific characteristics.

**Abstract:**

Associations between neurotransmitters, adrenergic receptor (ADR) mutations, and behaviors in chickens produced and domesticated by artificial selection remain unclear. This study investigates the association of neurotransmitters and ADR mutations with egg laying and cockfighting—behaviors associated with significantly different breeding backgrounds—in Shaver Brown and Shamo chickens. Accordingly, the whole sequences of nine ADR genes were determined, and nine amino acid-specific mutation sites from five genes (ADRα1A: S365G, ADRα1D: T440N, ADRα2A: D273E, ADRβ1: N443S, S445N, ADRβ3: R342C, Q404L, and P406S) were extracted. Evolutionary analysis showed that these mutations were not ancestrally derived. These results confirm that the mutations at these sites were artificially selected for domestication and are breed specific. *N_ST_* population analysis confirmed a difference in the degree of genetic differentiation between the two populations in seven genes. The results further confirm differences in the degree of genetic differentiation between the two populations in Shaver Brown (*ADRA1B* and *ADRA1D*) and Shamo (*ADRA1A* and *ADRA2B*) chickens, indicating that the ADR gene differs between the two breeds. The effects of artificial selection, guided by the human-driven selection of desirable traits, are reflected in adrenaline gene mutations. Furthermore, certain gene mutations may affect domestication, while others may affect other traits in populations or individuals.

## 1. Introduction

Associations between neurotransmitters, adrenergic receptor (ADR) mutations, and behaviors in modern chickens produced and domesticated by artificial selection remain unclear. The effects of neurotransmitters and adrenergic receptor (ADR) mutations may be related to the behavior of Shamo chickens, bred for cockfighting, and Shaver Brown chickens, bred for egg-laying [1]. 

The brains of domesticated chickens have been used to study the relationship between monoamine neuron concentrations, aggression, and polymorphism in nine ADR genes [1]. The analysis of monoamines showed significantly higher (*p* = 0.0087) noradrenaline levels in the Shamo midbrain than in the Shaver Brown midbrain, suggesting that noradrenaline may be associated with the fighting spirit of gamecocks. Brain monoamines include noradrenaline, adrenaline, dopamine, and serotonin [2,3,4,5]. Most monoamine neuron groups are located in the brainstem and control a large area of the brain [6,7,8]. Furthermore, monoamines released from the nerve terminal exert their actions via each monoamine receptor, and some are returned to the nerve terminal via carrier transport [9,10,11,12]. Among the monoamines that play an important role in neurotransmission, only noradrenalin appears to show the most significant difference in terms of concentration between Shamo and Shaver Brown chickens [1]. 

The genetic analysis of adrenergic receptors has also revealed specific mutations in *ADR β2* (T44I and Q232R). This analysis confirmed that the highly aggressive gamecocks have a significantly higher noradrenaline concentration than domesticated chickens and have specific *ADRβ2* gene mutations [1]. In addition, evolutionary tree analysis of Phasianidae did not show these mutations in the ancestral species, suggesting that the fighting behavior of gamecocks was inherited from artificial selection and that these genes affect the domestication of chickens [1]. However, the link between domestication and ADR gene mutations produced by artificial selection in either breed has not yet been clarified. Therefore, this study builds on our previous study and reports the effects of ADR gene mutations in Shamo and Shaver Brown chickens. 

In this study, we investigate the association of neurotransmitters and ADR with egg laying and cockfighting in Shaver Brown and Shamo chickens, which are behaviors associated with significantly different breeding backgrounds. The results of this study demonstrate the influence of artificial selection in closely related species and confirm its effect on the phenotypes of these breeds.

## 2. Materials and Methods

### 2.1. Chicken Samples

Samples from three Shaver Brown chickens (Shaver Brown: N5, N6, and N7) and three gamecocks (Shamo: S6, S7, and S9), which had been improved through artificial selection in cultural environments with different purposes, were used in this study (Figure 1). These samples were collected during our previous investigation of neurotransmitter concentrations in the brain (24 weeks after birth). Significantly higher adrenaline concentrations were observed in the three Shamo chickens, whereas Shaver Brown chickens had significantly lower adrenaline concentrations [1]. DNA was extracted from the blood and sequenced, and the sequences were submitted to the International Nucleotide Sequence Database Collaboration (INSDC). The primers are shown in Appendix A. Table 1 lists the sample accession numbers, including Red Jungle Fowl data from previous research on dopamine and aCGH analyses [13]. Female chickens were used because they are invaluable for the genetic improvement of chicken lines. For example, in Shamo, Japan, breeding stock of these valuable species has never been interbred with other species; moreover, breeders almost never trade their breeding stock of females with each other [1,14,15,16]. All animal experiments were conducted in strict compliance with the ethical guidelines of Tokai University, Japan. The experimental protocol was approved by the Animal Investigation Committee of Tokai University, Japan (Approval Nos. 141024 and 152010) [1].

Shamo chickens have a strong fighting spirit and have been used for cockfighting for over 1000 years [2]. However, due to their strong fighting spirit, it is difficult to breed them together with other chickens, and they must be kept in separate cages [14]. Shaver Brown chickens can be bred in flocks [17,18].

### 2.2. Molecular Phylogeny Analysis of the Complete Nine ADR Genes

A phylogenetic tree was constructed using the unweighted pair group method with arithmetic means (UPGMA). The UPGMA algorithms were incorporated into CLUSTALW-MEGA var. X using distances corrected for multiple hits based on Kimura’s two-parameter model [14,15,19,20]. The phylogenetic trees used a bootstrap analysis of 1000 replications to assess the statistical confidence in the branching order of the trees [14,15]. Evolutionary analysis was performed on the nine ADR genes of Shaver Brown, Shamo, *Gallus gallus gallus*, *Coturnix japonica*, *Meleagris gallopavo*, *Numida meleagris*, and *Phasianus colchicus* (Appendix A). The approximate lengths of the nine ADR genes for evolutionary analysis were as follows: *ADRA1A*, 1400 bp; *ADRA1B*, 1520 bp; *ADRA1D*, 1530 bp; *ADRA2A*, 1330 bp; *ADRA2B*, 1035 bp, *ADRA2C*; 1340bp, *ADRB1*;1430 bp; *ADRB2*, 1190 bp; and *ADRB3*, 1310 bp. Sites representing gaps in any of the aligned sequences were excluded from the analysis. In addition, a BLAST search at NCBI was performed to confirm mutation sites other than those in the wild birds.

### 2.3. Analysis of Mutation Site Location Using WoLF PSORT and TMHMM

WoLF PSORT (https://wolfpsort.hgc.jp/, accessed on 29 October 2022) was used to predict the subcellular localization sites of proteins based on their amino acid sequences. TMHMM-2.0 (https://services.healthtech.dtu.dk/service.php?TMHMM-2.0, accessed on 30 September 2022) is a predictive tool for the identification of protein transmembrane helices. Using these tools, it was previously confirmed that ADR is an intracellular 7-transmembrane protein [1,16].

### 2.4. Index of Nucleotide Differentiation (N_ST_) Analysis of Adrenaline Receptor Genes (Fixation Index)

From the previous study, *N_ST_* analyses for the dopamine D4 receptor (DRD4) genes of the three domesticated chicken populations were performed to characterize the evolutionary differentiation of the genes in the three populations. The *N_ST_* value of DRD4 for Shamo was distinctly larger than the other domesticated chicken populations [16,21]. 

To measure the degree of evolutionary differentiation between the two domesticated chicken groups, the fixation index (*N_ST_*) for each of the nine genes was calculated [16]. In addition, an estimate of the heterozygosity per site (R; purine, Y; pyrimidine, M; amino, K; keto, S; strong interaction, and W; weak interaction) was used to calculate the *N_ST_* (Appendix A) [21,22]. The details are described in Komiyama et al. (2014).

The *N_STij_* for the *i*-th and *j*-th subpopulation is defined as:$$N_{STij}=\frac{H_{Ti}-H_{Sij}}{H_{Ti}}\;\;\;$$
in which
$$H_{Ti}=1-\left(P_{Ai}^{2}+P_{Ti}^{2}+P_{Gi}^{2}+P_{Ci}^{2}\right)$$
and
$$H_{Sij}=1-\left(P_{Aij}^{2}+P_{Tij}^{2}+P_{Gij}^{2}+P_{Cij}^{2}\right)\;N_{STj}=\overset{n}{\underset{i=1}{\sum }}\frac{N_{STij}}{n}\;\;\;\;\;$$
for *j* = 1 to *m*.

However, the absolute values of the negative values in the present case are so small that it can be safely assumed that they are not significantly different from zero. In addition, the *N_ST_* can be used not only for coding regions, but also for noncoding regions of the genomes in question when nucleotide sites are segregated. If this method is applied to introns or noncoding regions in which the evolutionary rate is generally higher than that of exons or RNA coding regions, the evolutionary differentiation of closely related populations, such as the present chicken populations, can be studied. 

## 3. Results

### 3.1. Sequence Analysis of the Nine ADR Subtypes in the Two Breeds

First, the complete gene sequences of the nine adrenergic receptor subtypes, *ADRA1A*, *ADRA1B*, *ADRA1D*, *ADRA2A*, *ADRA2B*, *ADRA2C*, *ADRB1*, *ADRB2*, and *ADRB3*, were determined by direct sequencing.

The results show that there are four unique mutation sites in Shaver Brown chickens, thirteen unique mutation sites in Shamo chickens, and seven unique mutation sites that have common sites between the two species (Table 2 and Appendix A). Two mutation sites were in noncoding regions. A total of 24 specific gene mutation sites were identified. Four unique gene mutation sites were identified in Shaver Brown chickens: ADRα1A (S365G), ADRα1B (R258Q, V494A), and ADRβ1 (G444S). The unique mutation sites in Shamo chickens are ADRα1D (L58W, T440N), ADRα2A (V296I), ADRα2B (R138Q, R210H, V292M), ADRβ1 (N443S, S445N, R466C), and ADRβ2 (A15T, T44I, Q232R, T277M). The common mutation sites are ADRα2A (V58I, D273E), ADRβ1 (R403Q), and ADRβ3 (R342C, S396P, Q404L, and P406S). In addition, 23 of these mutations were heterozygous (Table 2 and Appendix A). By analyzing these two breeds, it was possible to divide the mutations into three groups based on the nine types of ADR genes. Five mutation sites of ADRβ1 were confirmed among the three groups: Shaver Brown, Shamo, and both breeds (Table 2). Orange indicates Shaver Brown-specific mutations, whereas blue indicates Shamo-specific mutations. Green indicates mutations common to both breeds. No color indicates external mutations (Appendix A).

### 3.2. Prediction of Transmembrane Helices in ADR Proteins

Gene locations in membrane proteins were investigated by WoLF PSORT analysis (Appendix A). Each mutation in the proteins encoded by the nine ADR genes was predicted in the integral membrane domains by TMHMM (Appendix A). Only two mutations (ADRα2B, V292M; ADRβ2, A15T) were in the extracellular region and did not affect gene function. Other mutations were confirmed in the TMhelix and intracellular region (Appendix A). Subsequently, 24 specific amino acid mutation sites from the TMHMM analysis confirmed that the mutations were located in the transmembrane helices (Table 2). 

### 3.3. Identification of Mutation Sites and Evolution-Related Analysis of ADR Gene Mutation Sites with Galliformes 

Next, an evolutionary analysis of the 24 specific mutations was conducted. For the identification of mutation sites and evolutionary analysis, the sequences of each ADR gene in Galliformes (*Gallus gallus gallus*, *Coturnix japonica*, *Meleagris gallopavo*, *Numida meleagris*, and *Phasianus colchicus*) in the DDBJ/EMBL-EBI/GenBank databases (Appendix A) were analyzed. These samples were selected based on previous reports [1,23,24,25]. The presence or absence of mutation sites was confirmed by performing a BLAST search and alignment with the Galliformes sequences registered in the database. Eight specific genes were identified in both breeds (Table 3).

### 3.4. Evolution-Related Analysis of S365G Mutation Sites (ADRα1A) in Shaver Brown Chickens

The evolutionary pathways of these genes were confirmed using phylogenetic tree analysis. Four unique gene mutations were identified in Shaver Brown chickens: ADRα1A (S365G), ADRα1B (R258Q, V494A), and ADRβ1 (G444S). ADRα1A (S365G) was detected in three Shaver Brown chickens. None of the mutations could be identified in the ancestral pheasant or turkey species (Figure 2). ADRα1B (R258Q, V494A) was only present in individuals N6 and N7. ADRβ1 (G444S) was found only in N6 and was also observed in pheasants. Only ADRα1A (S365G) is considered to have been mutated by human mating and selection for generations, and as domestication progressed, the wild species sites mutated. This artificial selection resulted in the generation of mutations for Shaver Brown chickens. Then, these mutations have been conserved until now. Approximately all mutations occurred in heterozygous sites (R: purine).

### 3.5. Evolution-Related Analysis of Three Mutation Sites: T440N (ADRα1D), S443N, and N445S (ADRβ1) in Shamo Chickens

There are 13 unique mutation sites in Shamo chickens: ADRα1D (L58W, T440N), ADRα2A (V296I), ADRα2B (R138Q, R210H, V292M), ADRβ1 (S443N, N445S, R466C), and ADRβ2 (A15T, T44I, Q232R, and T277M). ADRβ2 (T44I, Q232R, and R466C) mutations are heritable from Galliformes [1]. ADRα1D (L58W), ADRα2A (V296I), and ADRα2B (R210H) were observed in only one Shamo chicken sample. ADRβ1 (R466C) was also identified in both pheasants and turkeys. ADRα2B (V292M) and ADRβ2 (A15T) were outside the region based on TMHMM. Mutations in ADRα1D (T440N) and ADRβ1 (S443N, N445S) were confirmed in the three Shamo chickens (Figure 3).

### 3.6. Evolution-Related Analysis of Four Common Mutation Sites: D273E (ADRα1A), S443N, and N445S (ADRβ1) in Shaver Brown and Shamo Chickens

ADRα2A (V58I, D273E), ADRβ1 (R403Q), and ADRβ3 (R342C, S396P, Q404L, P406S) were common mutations specific to the two breeds. Only S396P of ADRβ3 was also found in pheasants and Japanese quails. ADRα2A (V58I) was confirmed in pheasants and ADRβ1 (R403Q) in guinea fowl. Therefore, these three mutations were identified in two of the domesticated chickens, suggesting that they were strongly influenced by artificial selection (Figure 4).

### 3.7. N_ST_ Analysis

The samples for the analysis of the nine ADR genes were obtained from three Shaver Brown chickens (N5, N6, and N7) and three Shamo chickens (S6, S7, and S9) (Table 1). Table 4 shows the total number of base pairs sequenced for the nine ADR genes and their total segregating sites. Most segregation sites were confirmed to be heterozygous sites. *N_ST_* analysis is an effective analysis method for heterozygous sites.

Next, an *N_ST_* population analysis was performed for the Shaver Brown and Shamo breeds. Our results were as follows: *ADRA1A*, 0.007; *ADRA1B*, 0.031; *ADRA1D*, 0.064; *ADRA2A*, 0.006; *ADRA2B*, 0.075; *ADRB1*, 0.009; and *ADRB2*, 0.007 (Table 5 and Appendix A). These data confirm the difference in the degree of genetic differentiation between Shaver Brown and Shamo chickens. In particular, the occurrence of *ADRA1B*, *ADRA1D*, and *ADRA2B* is distinctly greater than that of other ADR genes. In addition, there are no differences in the occurrences of *ADRA2C* and *ADRB3* between the two breeds. Both are considered genes that are not influenced by artificial selection. Thus, there is a difference in the ADR gene expression between the two breeds. Table 4 shows the mutation rate of the segregating sites (%) in the seven ADR genes. *ADRA1A, ADRA2A*, and *ADRA2B* have higher rates than the other genes. However, *N_ST_* rates of the *ADRA1D* (0.064) gene were the most affected by artificial selection.

## 4. Discussion

The association between neurotransmitters, adrenergic receptors, mutations, and behaviors in chickens produced and domesticated by artificial selection has not yet been clarified. Changes in the site of a gene causes changes in amino acids, resulting in changes in proteins. Even with the same gene, there is a large difference in the phenotype depending on the mutation site and type of change. The occurrence of these mutations has a significant influence on chicken characteristics. Therefore, this study used Shaver Brown and Shamo chickens, which have significantly different breeding backgrounds for food and cockfighting, respectively. 

### 4.1. N_ST_ of the Shaver Brown and Shamo Chickens

First, nine complete ADR sequences were collected from the database. As the first analysis, to clarify genetic differentiation, the *N_ST_* was measured. The *N_ST_* population analysis showed that the greatest difference in the degree of genetic differentiation was between the following seven ADR genes: *ADRA1A*; 0.007, *ADRA1B*; 0.031, *ADRA1D*; 0.064, *ADRA2A*; 0.006, *ADRA2B*; 0.027, *ADRB1*; 0.009, and *ADRB2*; 0.007. In particular, *ADRA1B*, *ADRA1D*, and *ADRA2B* were found to occur more frequently than the other ADR genes. An analysis of heterosites revealed clear differences in the *N_ST_* values of the ADR genes. However, in *ADRA2C* and *ADRB3*, the *N_ST_* value did not change. This is because these two genes were not affected by artificial selection. On the other hand, seven ADR genetic differentiations among the two breeds were shown to be strongly related to the life characteristics of these breeds as a result of different purposes of artificial selection. 

### 4.2. Amino Acid Mutation Sites of Shaver Brown and Shamo Chickens

Based on the nine ADR gene sequence analyses, twenty-four specific amino acid mutation sites were found in eight ADR genes (Table 1). These results confirm the presence of three groups of different mutation sites in the two chicken breeds. The 24 specific amino acid mutation sites were confirmed using WoLF PSORT and TMHMM analyses. The results of the analyses show that twenty-two out of the twenty-four observed mutations in the eight ADR genes affect the function of membrane proteins. Some traits are associated with sites that were affected by artificial selection. This shows that ADR gene mutations are likely related to behavior (emotion) because they are found inside membrane proteins at most mutation sites [1,26,27,28,29,30]. This suggests a relationship between breed-specific behavior and these mutations. Here, the mutations present in all three individuals of each breed and the ADR genes shared by the two breeds were selected. The five ADR genes have eight mutation sites (Table 2). Mutations with possible individual differences were excluded from this study.

Evolutionary analysis was then performed using the eight sites of these genes to confirm whether these sites are breed specific. These eight sites could not be identified in the ancestral species Phesianus, Cotrunix, Meleagris, or Numida. Therefore, these mutations are considered the original mutations of Shaver Brown and Shamo and are thought to have been fixed in these breeds through artificial selection.

### 4.3. Effects of Genetic Mutation Sites of ADR in Shaver Brown and Shamo Chickens

ADRs are currently classified into three types—ADRα1, ADRα2, and ADRβ—and three subtypes [31]. ADRα1 (α1A, α1B, and α1D) is involved in vasoconstriction, pupil dilation, sleep, and prostate contraction [32,33,34,35,36,37,38]. ADRα2 (α2A, α2B, and α2C) is involved in various nervous system actions in addition to platelet aggregation and lipolysis suppression [39,40,41,42]. ADRβ1, mainly present in the heart, is involved in increased systolic forces, relaxation of the uterine smooth muscle, and activation of lipolysis [43,44,45]. ADRβ2, existing in bronchi and blood vessels, relaxes various smooth muscles, such as the bronchial smooth muscle, vascular smooth muscles (muscle and liver), and the uterine smooth muscle, as well as controls glucose metabolism [46,47,48,49,50,51,52]. ADRβ3 is present in adipocytes, the gastrointestinal tract, liver, and skeletal muscles and is expected to be present in the postsynaptic membrane of adrenergic nerves [53,54,55,56]; it is also known to affect basal metabolism [57]. 

Based on these results, ADRα1A and ADRα1B mutations were confirmed in Shaver Brown chickens. In addition, the *N_ST_* values were large for ADRα1B and ADRα1D. Shamo chickens have mutations in ADRα1D, ADRα2A, ADRα2B, ADRβ1, and ADRβ2. The *N_ST_* values were high for ADRα1A and ADRα2B. In addition, both breeds have common mutations in ADRα2A, ADRβ1, and ADRβ3.

In our previous study, we found differences in the concentrations of neurotransmitters in the brain, especially noradrenaline [1]. Through each receptor, it can be speculated that the genes that affect vasoconstriction, smooth muscles, and the action of the digestive system were altered by these different neurotransmitter concentrations. The genetic mutation sites in the adrenergic receptor of Shaver Brown and Shamo chickens resulting from artificial selection affect vasoconstriction, smooth muscles, and the action of the digestive system. Gene site mutations have been used as criteria for individual selection with a lack of knowledge of the underlying mechanism by humans and have thus accumulated in specific breeds through artificial selection. 

These ADR gene mutations may help elucidate the causes of vascular, gastrointestinal, and cardiovascular diseases [58,59,60,61,62,63,64,65,66,67]. Furthermore, they may also assist in elucidating the roles of neurotransmitters in the human brain and their participation in stress-related conditions, such as panic disorder, depression, syncope, and anxiety [68,69,70,71,72,73]. 

This study will be useful for investigating the relationship between the onset of conditions that affect vasoconstriction, smooth muscles, and the function of the digestive system as a result of narrow-range mating.

## 5. Conclusions

Nine types of adrenaline gene sequences were analyzed, and the identified twenty-four specific mutation sites were the result of artificial selection. *N_ST_* population analysis confirmed that there is a difference in the degree of genetic differentiation between *ADRA1A*, *ADRA1B*, *ADRA1D*, and *ADRA2B*. Moreover, these results confirmed that the eight mutation sites are expected to be affected by artificial selection. Evolutionary analysis indicated the absence of an ancestral mutation. These results confirmed that the mutations at these sites were artificially selected for domestication and are breed specific.

## Figures and Tables

**Figure 1 biology-12-00169-f001:**
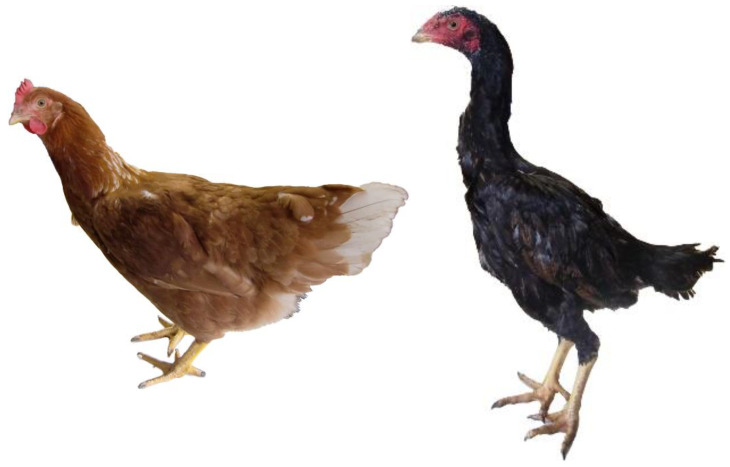
Characteristics of female Shaver Brown and Shamo chickens. Left: Shaver Brown. Right: Shamo [1].

**Figure 2 biology-12-00169-f002:**
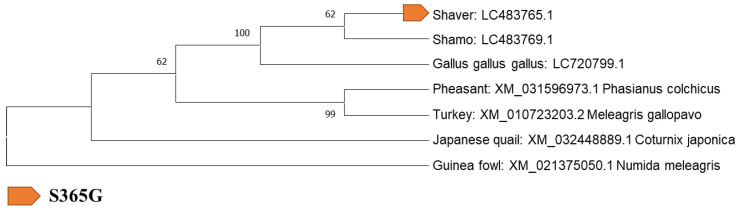
Phylogenetic analysis of the adrenergic receptor (ADR) gene ADR*α*1A.

**Figure 3 biology-12-00169-f003:**
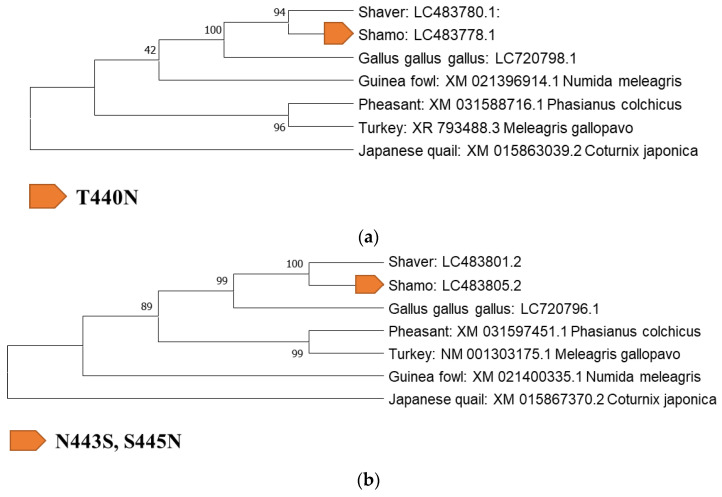
Phylogenetic analysis of the adrenergic receptor (ADR) genes *ADRα1D* (**a**) and *ADR β1* (**b**).

**Figure 4 biology-12-00169-f004:**
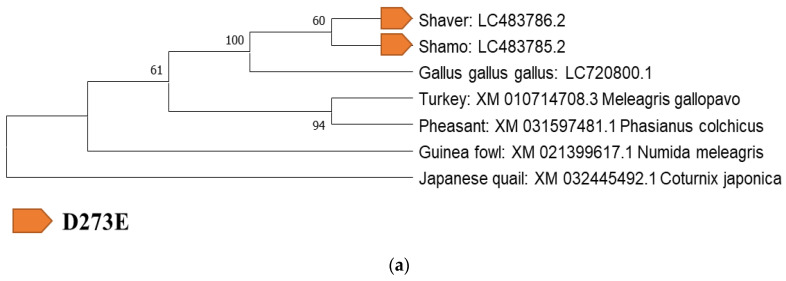
Phylogenetic analysis of the adrenergic receptor (ADR) genes *ADRα2A* (**a**) and *ADRβ3* (**b**).

**Table 1 biology-12-00169-t001:** Accession numbers of nine adrenergic receptor (ADR) genes in Shaver Brown and Shamo chickens.

Chicken Breed	Sample No.	Gene Name								
		*ADRA1A*	*ADRA1B*	*ADRA1D*	*ADRA2A*	*ADRA2B*	*ADRA2C*	*ADRB1*	*ADRB2*	*ADRB3*
Shaver Brown	N5	LC483765.1	LC483771.1	LC483780.1	LC483786.2	LC483789.1	LC483795.1	LC483801.2	LC483810.1	LC483813.1
	N6	LC483766.1	LC483772.1	LC483781.1	LC483787.2	LC483790.1	LC483796.1	LC483802.2	LC483811.1	LC483814.1
	N7	LC483767.1	LC483773.1	LC483782.1	LC483788.2	LC483791.1	LC483797.1	LC483803.2	LC483812.1	LC483815.1
Shamo	S6	LC483768.1	LC483774.1	LC483777.1	LC483783.2	LC483792.1	LC483798.1	LC483804.2	LC483807.1	LC483816.1
	S7	LC483769.1	LC483775.1	LC483778.1	LC483784.2	LC483793.1	LC483799.1	LC483805.2	LC483808.1	LC483817.1
	S9	LC483770.1	LC483776.1	LC483779.1	LC483785.2	LC483794.1	LC483800.1	LC483806.2	LC483809.1	LC483818.1
Red Jungle Fowl	222	LC720799.1	LC720797.1	LC720798.1	LC720800.1	LC720804.1	LC720803.1	LC720796.1	LC720802.1	LC720801.1

**Table 2 biology-12-00169-t002:** The 24 amino acid mutation sites in the nine adrenergic receptor (ADR) genes in Shamo and Shaver Brown chickens.

Chicken Breed		Gene Name									Total Mutation Number
		ADRα1A	ADRα1B	ADRα1D	ADRα2A	ADRα2B	ADRα2C	ADRβ1	ADRβ2	ADRβ3	
1	Shaver Brown	S365G	R258Q V494A	—	—	—	—	G444S	—	—	4
2	Shamo	—	—	L58W T440N	V296I	R138Q R210H V292M	—	N443S S445N R466C	A15T T44I Q232R T277M	—	13
3	Both breeds	—	—	—	V58I D273E	—	—	R403Q	—	R342C S396P Q404L P406S	7

A, alanine; C, cysteine; D, aspartic acid; E, glutamic acid; G, glycine; H, histidine; I, isoleucine; L, leucine; M, methionine; N, asparagine; P, proline; Q, glutamine; R, arginine; S, serine; T, threonine; V, valine.

**Table 3 biology-12-00169-t003:** The eight amino acid mutation sites in the five adrenergic receptor (ADR) genes in Shamo and Shaver Brown chickens.

	Chicken Breed	Gene Name				
		ADRα1A	ADRα1D	ADRα2A	ADRβ1	ADRβ3
1	Shaver Brown	S365G	—	—	—	—
2	Shamo	—	T440N	—	N443S, S445N	—
3	Both breeds	—	—	D273E	—	R342C, Q404L, P406S

S, serine; G, glycine; T, threonine; N, asparagine; D, aspartic acid; E, glutamic acid; R, arginine; C, cysteine; Q, glutamine, L, leucine; P, proline.

**Table 4 biology-12-00169-t004:** The total number of base pairs sequenced for the nine adrenergic receptor (ADR) genes and their total segregating sites for the index of nucleotide differentiation (*N_ST_*) analysis.

Chicken Breed	Gene Name								
	*ADRA1A*	*ADRA1B*	*ADRA1D*	*ADRA2A*	*ADRA2B*	*ADRA2C*	*ADRB1*	*ADRB2*	*ADRB3*
Shaver Brown and Shamo (bp)	1404	1524	1536	1335	1038	1341	1434	1194	1314
Segregating sites and mutation rate (%)	14 (1.0)	9 (0.59)	7 (0.46)	22 (1.65)	10 (0.96)	1 (0.07)	8 (0.56)	9 (0.75)	9 (0.68)
Heterozygous sites	14	6	6	22	9	1	6	9	9

**Table 5 biology-12-00169-t005:** Index of nucleotide differentiation (*N_ST_*) analysis for the nine adrenergic receptor (ADR) genes between the two chicken breeds.

Chicken Breed	Gene Name								
	*ADRA1A*	*ADRA1B*	*ADRA1D*	*ADRA2A*	*ADRA2B*	*ADRA2C*	*ADRB1*	*ADRB2*	*ADRB3*
Shamo	0.017	0.006	−0.027	−0.004	0.027	0	−0.003	−0.007	0
Shaver Brown	0.001	0.031	0.064	0.006	−0.009	0	0.009	0.007	0

## Data Availability

The data presented in this study are available in the article and Appendix A.

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
