# Peer review of "Effects of Genetic Mutation Sites in ADR Genes on Modern Chickens Produced and Domesticated by Artificial Selection"

_biology, 2023, doi:10.3390/biology12020169_

Round 1
Reviewer 1 Report
The authors previously proposed the potential impact of neurotransmitters and adrenergic receptor (ADR) mutations on the behavior of Shamo chickens bred for cockfighting vs. Shaver Brown chickens bred for egg laying. In this manuscript, the author examined nine types of full adrenergic receptor (ADR) gene sequences, and noticed that 24 particular mutation sites were the consequence of Shaver Brown and Shamo chicken selection. The study segregated Shamo-specific mutations, Shaver Brown-specific mutations, and mutations common to both breeds into three categories. The findings revealed that artificial selection would have an effect on eight specific sites. Furthermore, the detected mutations were not ancestral, according to an evolutionary study. Based on these findings, the author suggested that the eight mutations at these locations were artificially selected for domestication and breed-specificity. This shows that various mutations may have distinct impacts on vasoconstriction, smooth muscle, and digestive system activity, depending on the breed's specific function.
This manuscript is clearly written. The data is well analyzed, although the number of analyzed sequences and breeds is limited likely due to the availability of samples. The discussion is speculative because it is hard to analyse the effect of altered amino acids on the behavior of adult chickens. It would be nice to carry out some cell biological and pharmacological assays using cultured cells (eg, transfection of mutated receptors). However, I think this manuscript itself is valuable. I have several suggestions.
1) According to the Chicken Gene Nomenclature Consortium (http://birdgenenames.org/cgnc/), gene symbols are written in upper case letters. eg, ADRα1A -> ADRA1A, ADRβ1 -> ADRB1. Gene symbols are italicized. This rule is widely used in many chicken papers (but not always). If amino acid residues are shown, the name refers to "protein" not "gene" (eg, line 163-168, ADRα1A (S365G) ). In this manuscript, the author italicizes most protein names and uses them inconsistently throughout the text.
My suggestion would be:
Gene symbol, ADRA1A (Italic)
Protein name, ADRα1A or ADRA1A (not italicized)
2) Line 306-308. " This showed that ADR gene mutations are likely related to behavior (emotion) because they are found inside membrane proteins at most mutation sites [1,26-30]. This suggests a relationship between breed-specific behavior and these mutations. "
It is unclear how "inside membrane proteins at most mutation sites" is related to behavior (emotion).
Related to this, the author used PSORT and TMHMM to predict the mutated sites. Did the author look at AlphaFold 2 (Google's Deep learning prediction)? eg. https://alphafold.ebi.ac.uk/entry/F1NP52 (Chicken ADRA1A structure).
3) Although the materials were used in the author's previous study, this kind of ethical statement (from reference #1) should be included in the method or in the last statement.
"All animal experiments were conducted in strict compliance with the ethical guidelines of Tokai University, Japan. Approval was also obtained from the Animal Investigation Committee of Tokai University, Japan (Approval Nos. 141024 and 152010)."
Author Response
Dear Reviewer 1,
I would like to take this opportunity to thank you and the other reviewers for your pertinent and helpful comments. I have provided point-by-point responses to each comment below and believe that my manuscript has been significantly strengthened as a result of the revisions.
I appreciate your continued guidance and feedback regarding my manuscript and hope that you will now find it acceptable for publication in your journal.
Tomoyoshi Komiyama
Reviewer 1
Comments and Suggestions for Authors
The authors previously proposed the potential impact of neurotransmitters and adrenergic receptor (ADR) mutations on the behavior of Shamo chickens bred for cockfighting vs. Shaver Brown chickens bred for egg laying. In this manuscript, the author examined nine types of full adrenergic receptor (ADR) gene sequences, and noticed that 24 particular mutation sites were the consequence of Shaver Brown and Shamo chicken selection. The study segregated Shamo-specific mutations, Shaver Brown-specific mutations, and mutations common to both breeds into three categories. The findings revealed that artificial selection would have an effect on eight specific sites. Furthermore, the detected mutations were not ancestral, according to an evolutionary study. Based on these findings, the author suggested that the eight mutations at these locations were artificially selected for domestication and breed-specificity. This shows that various mutations may have distinct impacts on vasoconstriction, smooth muscle, and digestive system activity, depending on the breed's specific function.
This manuscript is clearly written. The data is well analyzed, although the number of analyzed sequences and breeds is limited likely due to the availability of samples. The discussion is speculative because it is hard to analyse the effect of altered amino acids on the behavior of adult chickens. It would be nice to carry out some cell biological and pharmacological assays using cultured cells (eg, transfection of mutated receptors). However, I think this manuscript itself is valuable. I have several suggestions.
1) According to the Chicken Gene Nomenclature Consortium (http://birdgenenames.org/cgnc/), gene symbols are written in upper case letters. eg, ADRα1A -> ADRA1A, ADRβ1 -> ADRB1. Gene symbols are italicized. This rule is widely used in many chicken papers (but not always). If amino acid residues are shown, the name refers to "protein" not "gene" (eg, line 163-168, ADRα1A (S365G) ). In this manuscript, the author italicizes most protein names and uses them inconsistently throughout the text.
My suggestion would be:
Gene symbol, ADRA1A (Italic)
Protein name, ADRα1A or ADRA1A (not italicized)
Reply: Thank you for the suggestion, which I agree with. Accordingly, I have changed style of the gene and protein symbols.
2) Line 306-308. " This showed that ADR gene mutations are likely related to behavior (emotion) because they are found inside membrane proteins at most mutation sites [1,26-30]. This suggests a relationship between breed-specific behavior and these mutations. "
It is unclear how "inside membrane proteins at most mutation sites" is related to behavior (emotion).
Reply: Thank you for the suggestion, and I agree with you. In my previous study, I measured neurotransmitter levels in both chicken breeds and confirmed a significant difference in the amount of noradrenaline. I was also able to confirm differences in adrenergic receptors between the two breeds, which correlated with differences in genetic background and behavior in the two breeds.
To provide further evidence for this relationship, I am currently performing an array scan experiment by adding noradrenaline to chicken, mouse, and human smooth muscle cells. Results of this experiment will be reported in the future.
Related to this, the author used PSORT and TMHMM to predict the mutated sites. Did the author look at AlphaFold 2 (Google's Deep learning prediction)? eg. https://alphafold.ebi.ac.uk/entry/F1NP52 (Chicken ADRA1A structure).
Reply: Thank you for the pertinent suggestion. Here, I did not consult AlphaFold 2. Nonetheless, I would like to study AlphaFold 2, analyze each reported study, and use it for my next study. As can be seen from the NCBI database, I believed that it would be difficult to construct the structure of a membrane protein such as ADR. Therefore, I used PSORT and TMHMM, as in my previous paper, to check only the position of the protein.
3) Although the materials were used in the author's previous study, this kind of ethical statement (from reference #1) should be included in the method or in the last statement.
"All animal experiments were conducted in strict compliance with the ethical guidelines of Tokai University, Japan. Approval was also obtained from the Animal Investigation Committee of Tokai University, Japan (Approval Nos. 141024 and 152010)."
Reply: Thank you for the suggestion, which I agree with. Accordingly, I have added a statement on ethical approval to the section “2.1. Chicken samples”.
Also, some sentences were revised, I have identified them with yellow highlights. Please check them again.
This manuscript checked by a professional English editing service.

Reviewer 2 Report
This draft compared 9 ADR genes sequenced in 3 egg laying chickens and 3 gamecock chickens, anlalyzed the locations of the mutations in protein and checked the existence of these mutations in Red Jungle Fowl, etc.The introduction part overestimated the content of the study. Compared with previous studies, there were no new conclusions about ADR genes, nor revealed the behavior of fighting cocks or laying eggs or something else. The results part was not clear enough and the discussion and conclusion sections were too arbitrary with insufficient information. This paper was more like a part work of a study.
line 163, four unique mutations, but in line 164, there were four .
line 166, three mutations were not the same as in table 2.
line 170-171, five mutations were confirmed among the three groups. what is the meaning? which five? which three?
line 187, The location of gene function in membrane proteins was investigated. what's the meaning?
line 190, mutations V292M, A15T were in the intracellular regions. These two mutations were of 24 mutations. So, why in lines 192-193, 24 mutations were transmembrane?
...
Based on the results, I think it's safe to say that we've got mutations in 9 genes in 3 egg laying and 3 gamecocks individuals, some of which were common, and some of which were unique for these chickens, most of the sites were in the transmembrane region, and some of which were not present in Red Jungle Fowl. Given the function of ADR genes, we hypothesized that these mutations may be related to certain behaviors of chickens, nothing more. As to whether these variants were specific to the breed, they need to be tested in a large population, and we often come across mutations in several individuals that are common but not at the population level. As for the relationship between these mutations and behavior, it cannot be determined by the discovery of these mutations.
Author Response
Dear Reviewer 2:
I would like to take this opportunity to thank you and the other reviewers for your pertinent and helpful comments. I have provided point-by-point responses to each of your comments below and believe that the manuscript has significantly been strengthened as a result of the revisions.
I appreciate your continued guidance and feedback regarding my manuscript and hope that you will now find it acceptable for publication in your journal.
Tomoyoshi Komiyama
Reviewer 2
Comments and Suggestions for Authors
This draft compared 9 ADR genes sequenced in 3 egg laying chickens and 3 gamecock chickens, anlalyzed the locations of the mutations in protein and checked the existence of these mutations in Red Jungle Fowl, etc.The introduction part overestimated the content of the study. Compared with previous studies, there were no new conclusions about ADR genes, nor revealed the behavior of fighting cocks or laying eggs or something else. The results part was not clear enough and the discussion and conclusion sections were too arbitrary with insufficient information. This paper was more like a part work of a study.
line 163, four unique mutations, but in line 164, there were four.
Reply: Accordingly, the sentence has been revised to indicate that there were four mutations, as the R403Q site was deleted from ADRB1.
line 166, three mutations were not the same as in table 2.
Reply: The sentence has been revised to correspond to the results shown in Table 2. Furthermore, I have reconfirmed all ADRB1 sequences.
line 170-171, five mutations were confirmed among the three groups. what is the meaning? which five? which three?]
Reply: The sentence has been revised to correspond to the results in Table 2.
line 187, The location of gene function in membrane proteins was investigated. what's the meaning?
Reply: The sentence has been revised accordingly for clarity.
Gene location in membrane proteins was investigated by WoLF PSORT analysis (Table S3). Each mutation in the protein encoded by the nine ADR genes was predicted in the integral membrane domains by TMHMM (Table S4).
line 190, mutations V292M, A15T were in the intracellular regions. These two mutations were of 24 mutations. So, why in lines 192-193, 24 mutations were transmembrane?
Reply: The mutations ADRα2B, V292M, and ADRβ2, A15T were in the extracellular region. Therefore, in total, 24 specific amino acid mutation sites were present. This has been clarified in the corresponding sentence in the revised manuscript (lines 190–191).
Based on the results, I think it's safe to say that we've got mutations in 9 genes in 3 egg laying and 3 gamecocks individuals, some of which were common, and some of which were unique for these chickens, most of the sites were in the transmembrane region, and some of which were not present in Red Jungle Fowl. Given the function of ADR genes, we hypothesized that these mutations may be related to certain behaviors of chickens, nothing more. As to whether these variants were specific to the breed, they need to be tested in a large population, and we often come across mutations in several individuals that are common but not at the population level. As for the relationship between these mutations and behavior, it cannot be determined by the discovery of these mutations.
Reply: In my previous study, I measured neurotransmitter levels in both chicken breeds and confirmed a significant difference in the levels of noradrenaline. I was also able to confirm differences in adrenergic receptors between the two breeds, which correlated with differences in their genetic background and behaviors.
In addition, in this study, I have discussed the origin of the fighting spirit of the Shamo (gamecock) breed. This study extracts differences in adrenergic receptor mutations between the Shamo and Shaver browns. The identification of these mutations forms the basis for the next planned behavioral investigation in transgenic mice or chickens. As shown in the findings of the previous report, the levels of noradrenaline in the brain were clearly different between the two breeds; therefore, I believe that the differences in the mutations in these receptors correspond to the differences in breed characteristics.
Also, some sentences were revised, I have identified them with yellow highlights. Please check them again.
This manuscript checked by a professional English editing service.

Reviewer 3 Report
In this manuscript, the author surveys the genetic variations of nine adrenoceptors in two chicken breeds of distinct behaviour and morphologies. Substantial phylogenic and evolutionary analyses have been performed. This work has provided some suggestive insight into the neurobiological explanation of the behaviours of two chicken breeds but direct evidence is lacking. In addition, multiple issues listed below must be addressed:
(1) The method for the sequence identification of nine ADR genes is not well provided. Rather than stating “DNA was extracted from the blood, sequenced, and the sequences were submitted to the International Nucleotide Sequence Database Collaboration (INSDC)”, the related primers should be provided if any have been used.
(2) Lines 81-83 and the whole paragraph of 2.4 “Index of nucleotide differentiation (NST) analysis of adrenaline receptor genes (fixation index)” highly resemble sections in “Komiyama, Tomoyoshi, et al. "Dopamine receptor genes and evolutionary differentiation in the domestication of fighting cocks and long-crowing chickens." PLoS One 9.7 (2014): e101778” and must be carefully rephrased.
Line 96: Please explain more about the mtDNA in “Molecular phylogeny analysis of the complete mtDNA”.
Line 123: The wording and presentation should be improved:
for example:
“The results showed that the Shaver Brown ADR was distinctly larger than the 123 Shamo ADR” (lines 123-124).
Here I guess it should be the evolutionary differentiation of one breed ADR is "larger" than that of another.
“Then, 24 specific amino acid mutation sites from the analysis confirmed the transmembrane helices of the mutation by TMHMM analysis (Table 2).” (lines 192-193)
Here the mutation is located in the transmembrane helices. Similarly, the “membrane proteins” could be changed to membrane “motifs” or “domains”.
“The occurrence of these mutations has a significant influence on chicken functions.”( Line 281-282)
Please consider stating more specific functions to substitute “chicken functions”.
The author finds “eight mutation sites” in five ADR genes that are breed-specific and are speculated to be originated from artificial selection during domestication. However, “eight specific-site genes” in the Simple Summary (line 13) and “the eight site genes” (line 362) are contradictory to the statement of 5 “ADR genes” of eight sites (line 310). Nevertheless, more specific nomenclature should be used to improve clarity.
Author Response
Dear Reviewer 3,
I would like to take this opportunity to thank you and the other reviewers for your pertinent and helpful comments. I have provided point-by-point responses to each of your comments below and believe that the manuscript has significantly been strengthened as a result of the revisions.
I appreciate your continued guidance and feedback regarding my manuscript and hope that you will now find it acceptable for publication in your journal.
Tomoyoshi Komiyama
Reviewer 3
Comments and Suggestions for Authors
In this manuscript, the author surveys the genetic variations of nine adrenoceptors in two chicken breeds of distinct behaviour and morphologies. Substantial phylogenic and evolutionary analyses have been performed. This work has provided some suggestive insight into the neurobiological explanation of the behaviours of two chicken breeds but direct evidence is lacking. In addition, multiple issues listed below must be addressed:
- The method for the sequence identification of nine ADR genes is not well provided. Rather than stating “DNA was extracted from the blood, sequenced, and the sequences were submitted to the International Nucleotide Sequence Database Collaboration (INSDC)”, the related primers should be provided if any have been used.
Reply: As suggested, the primer sequences have been listed in Table S1 in Supplementary Materials.
- Lines 81-83 and the whole paragraph of 2.4 “Index of nucleotide differentiation (NST) analysis of adrenaline receptor genes (fixation index)” highly resemble sections in “Komiyama, Tomoyoshi, et al. "Dopamine receptor genes and evolutionary differentiation in the domestication of fighting cocks and long-crowing chickens." PLoS One
9.7 (2014): e101778” and must be carefully rephrased.
Reply: The sentence has been revised accordingly.
Line 96: Please explain more about the mtDNA in “Molecular phylogeny analysis of the complete mtDNA”.
Reply: I apologize for the oversight. The current study only discusses adrenergic receptors in chickens.
Line 123: The wording and presentation should be improved:
for example:
“The results showed that the Shaver Brown ADR was distinctly larger than the 123 Shamo ADR” (lines 123-124).
Here I guess it should be the evolutionary differentiation of one breed ADR is "larger" than that of another.
Reply: The sentence has been revised accordingly.
“Then, 24 specific amino acid mutation sites from the analysis confirmed the transmembrane helices of the mutation by TMHMM analysis (Table 2).” (lines 192-193)
Here the mutation is located in the transmembrane helices. Similarly, the “membrane proteins” could be changed to membrane “motifs” or “domains”.
Reply: The sentence has been revised accordingly. Furthermore, “membrane proteins” has been revised to “membrane domains.”
“The occurrence of these mutations has a significant influence on chicken functions.”( Line 281-282)
Please consider stating more specific functions to substitute “chicken functions”.
Reply: The sentence has been revised accordingly. Furthermore, “chicken functions” has been revised to “chicken characteristics.”
The author finds “eight mutation sites” in five ADR genes that are breed-specific and are speculated to be originated from artificial selection during domestication. However, “eight specific-site genes” in the Simple Summary (line 13) and “the eight site genes” (line 362) are contradictory to the statement of 5 “ADR genes” of eight sites (line 310). Nevertheless, more specific nomenclature should be used to improve clarity.
Reply: The sentence has been revised accordingly, and these contradictions have been corrected.
Also, some sentences were revised, I have identified them with yellow highlights. Please check them again. This manuscript checked by a professional English editing service.

Reviewer 4 Report
My Primary concerns are there are just 3 samples per breed and its too few to do any evolutionary analysis. There is no mention on the effective population size of this breed so 3 is too few to do any evolutionary analysis.
Moreover most of what is described in this study is a repeat of the authors previous paper (https://www.nature.com/articles/s41598-020-63961-1#Sec9).
Unlike the previous paper where they reported results of Nst analysis on just four mutations specific to Shamo, here they have also reported mutations found on shaver brown.
I am finding it hard to understand what is different between these two work to warrant a separate paper. The author has not sufficiently justified it.
Some minor comments:
Provide more information on these breeds and why was ADR investigated. At the moment first part of the introduction reads like results and discussion.
Provide a table with the primers used for amplifying different ADR genes ?
Section 2.2 :- did you sequence mtDNA ? or should it be ADR genes.
Lines 159-162:- "The results showed that there are four unique mutation sites in Shaver Brown chick
ens, eleven unique mutation sites in Shamo chickens, and seven unique mutation sites
that have common sites between the two species (Table 2 and Table S2). Two mutation
sites were in noncoding regions. A total of 24 specific gene mutation sites were identified."
Must be rephrased. Not clear what you mean by four uniqe mutation in shaver, then 11 unique mutation in shamo nad then 7 unique mutation that is common (if it is common how is it unique, unique to what ?).
Clearly list how many were unique to shamo and how were unique to shaver, and what was common.
In your previous paper you reported 54 mutations, now its 24 what happend to to the other 30 ?
Author Response
Dear Reviewer 4,
I would like to take this opportunity to thank you and the other reviewers for your pertinent and helpful comments. I have provided point-by-point responses to each of your comments below and believe that the manuscript has significantly been strengthened as a result of the revisions.
I appreciate your continued guidance and feedback regarding my manuscript and hope that you will now find it acceptable for publication in your journal.
Tomoyoshi Komiyama
Reviewer 4
Comments and Suggestions for Authors
My Primary concerns are there are just 3 samples per breed and its too few to do any evolutionary analysis. There is no mention on the effective population size of this breed so 3 is too few to do any evolutionary analysis.
Reply: In this study, the samples were derived from individuals wherein a clear and significant difference in brain noradrenaline levels was confirmed previously. To do so, I analyzed all adrenergic receptors in each chicken. In fact, in my research environment, it is challenging to investigate brain neurotransmitter levels in Shaver brown or Shamo samples of the same age simultaneously. Nonetheless, I believe that the limitation of having a small number of samples is addressed by population genetic analyses (NST).
Moreover most of what is described in this study is a repeat of the authors previous paper (https://www.nature.com/articles/s41598-020-63961-1#Sec9).
Unlike the previous paper where they reported results of Nst analysis on just four mutations specific to Shamo, here they have also reported mutations found on shaver brown.
I am finding it hard to understand what is different between these two work to warrant a separate paper. The author has not sufficiently justified it.
Reply: The findings in this study differ from those of the previous report in terms of the interpretation of the results and points of discussion. In the previous report, I focused on the origin of gamecock's (Shamo) fighting spirit. In this study, I updated ADRA2A and ADRB1 sequences and included more mutations. As a result of these analyses, I have observed differences in the mutation sites of Shamo and Shaver Brown breeds, which have been reported herein.
Some minor comments:
Provide more information on these breeds and why was ADR investigated. At the moment first part of the introduction reads like results and discussion.
Reply: I have accordingly revised the Introduction.
Provide a table with the primers used for amplifying different ADR genes ?
Reply: Accordingly, the primer sequences have been listed in Table S1 of the Supplementary Materials.
Section 2.2 :- did you sequence mtDNA ? or should it be ADR genes.
Reply: I apologize for this oversight. The current study only discusses the adrenergic receptors of chickens.
Lines 159-162:- "The results showed that there are four unique mutation sites in Shaver Brown chickens, eleven unique mutation sites in Shamo chickens, and seven unique mutation sites
that have common sites between the two species (Table 2 and Table S2).
Reply: I apologize for this oversight; the number of the unique mutation sites in Shamo was incorrectly reported. I have revised the number of unique mutation sites from 11 to 13 in this sentence.
Two mutation sites were in noncoding regions. A total of 24 specific gene mutation sites were identified."
Must be rephrased. Not clear what you mean by four uniqe mutation in shaver, then 11 unique mutation in shamo nad then 7 unique mutation that is common (if it is common how is it unique, unique to what ?).
Clearly list how many were unique to shamo and how were unique to shaver, and what was common.
Reply: “unique”, used in this context, refers to breed-specific variation, similar to previous reports and literature.
Therefore, this unique points to the mutation site. Shaver Brown had four mutation sites. Shamo has 13 mutation sites. And there are 7 common sites for the 2 breeds (Table 2).
In your previous paper you reported 54 mutations, now its 24 what happend to to the other 30 ?
Reply: The 54 mutation sites reported in the previous article include all mutation sites across breeds. The amino acid mutation site number is 22. However, the total number of mutation sites in the two breeds this time is 58, more than the last time. Amino acid mutation site numbers alone are 24. This is the result of updating and re-analyzing the sequences.
Previous analysis: Total mutation sites are 54; 22 are amino acid mutation sites.
This study: Total mutation sites are 58; 24 are amino acid mutation sites (Table 2).
Also, some sentences were revised, I have identified them with yellow highlights. Please check them again. This manuscript checked by a professional English editing service.

Round 2
Reviewer 3 Report
Thanks to the author's effort. All the issues have been addressed, I have no further concerns about this manuscript.
Reviewer 4 Report
Thanks